# Escape and avoidance learning in the earthworm *Eisenia hortensis*

W. Jeffrey Wilson, Nicole C. Ferrara, Amanda L. Blaker and Charisa E. Giddings

Department of Psychological Science & Neuroscience Program, Albion College, Albion, MI, USA

## ABSTRACT

Interest in instrumental learning in earthworms dates back to 1912 when Yerkes concluded that they can learn a spatial discrimination in a T-maze. Rosenkoetter and Boice determined in the 1970s that the "learning" that Yerkes observed was probably chemotaxis and not learning at all. We examined a different form of instrumental learning: the ability to learn both to escape and to avoid an aversive stimulus. Freely moving "master" worms could turn off an aversive white light by increasing their movement; the behavior of yoked controls had no effect on the light. We demonstrate that in as few as 12 trials the behavior of the master worms comes under the control of this contingency.

## INTRODUCTION

In many ways earthworms are little more than "the intestines of the earth," as Aristotle described them in his *Historia Animalium* (353 BCE/1910). They have limited sensory capabilities, at least by vertebrate standards, and their motor functions are limited largely to locomotion, sexual coupling, exploratory head and mouth movements, and rapid writhing escape movements. Yet behavioral scientists have been interested in their ability to learn since shortly after the advent of comparative psychology.

*Yerkes (1912)* examined the ability of earthworms to learn a T-maze. He worked with *Allolobophora foetida*, a type of worm commonly known as a manure worm and today renamed *Eisenia foetida*. Remarkably, although Yerkes alluded to the collection of many worms, his 1912 paper is devoted to one worm, "No. 2." The worm was rewarded with access to a dark, moist tube for making the correct choice in a spatial discrimination task; an incorrect choice resulted in contact with sandpaper and either a strong saline solution or an electric shock. Over the course of many days of training his worm showed evidence of learning, but it was far from compelling.

Yerkes noted that the worm might follow a mucous trail, but believed that this could not account for the occasional perfect series of trials. Since Yerkes' early work many studies examined this type of instrumental learning in worms with the similar rewards and punishment, most typically using *Lumbricus terrestris*, a large burrowing worm (c.f. *Swartz, 1929*; *Datta, 1962*; *Zellner, 1966*; *McManus & Wyers, 1979*). However, *Rosenkoetter & Boice (1975)* demonstrated that this T-maze performance was likely not learning at all. Their data suggested that Yerkes' had underestimated the importance of

Corresponding author
W. Jeffrey Wilson,
wjwilson@albion.edu

the mucous trail; when they used a new maze on each trial, eliminating the possibility of persistent cues on the maze, performance in a T-maze remained near chance. After Rosenkoetter and Boice, interest in instrumental learning in the worm waned.

We revisit the question of instrumental learning in the earthworm with an attempt to demonstrate escape and avoidance learning. Psychologists have long studied these behaviors in other animals, most commonly rodents, who generally learn these tasks quite quickly (*Campbell & Kraeling, 1953*; *Bower, Starr & Lazarovitz, 1965*; see *Mackintosh, 1974* or *Domjan, 2010* for reviews). Given the adaptive value inherent in learning a response that reduces or prevents exposure to aversive and thus potentially harmful stimuli, we expect to find that earthworms can engage in this learning as well.

Our interest in learning in this organism represents more than idle curiosity. The relatively simple nervous system (a chain of nearly identical ganglia interconnected by three long giant fibers; *Bullock, 1945*) lends itself to easy neurophysiological examination — action potentials in these giant fibers can even be recorded noninvasively with proper amplification if the animal is lying on conductive electrodes (*Drewes, Landa & McFall, 1978*; *Kladt, Hanslik & Heinzel, 2010*). Other invertebrates have served as useful tools in the analysis of neural control of behavior (c.f. *Krasne & Glanzman, 1995*, for a review of invertebrate learning research up until the mid 1990s, *Hawkins, Kandel & Bailey, 2006*, for a discussion of the deep understanding of neural mechanisms of learning that derived from work in *Aplysia*; *Rankin, 2004* for an overview of the many behavioral capabilities of *C. elegans*.) We believe that the earthworm might also serve as a viable and inexpensive animal model for studies of the neural bases of learning.

We will use vibratory and light stimuli, like those used by many who have demonstrated Pavlovian conditioning in *Lumbricus* (c.f. *Ratner & Miller, 1959*; *Herz, Peeke & Wyers, 1967*; *Abramson & Buckbee, 1995*; *Watanabe et al., 2005*). We begin by determining the suitability of the stimuli that we plan to use, then in Experiment 2 we assess learning.

# EXPERIMENT 1

## Materials & Methods

### Subjects

We worked with the epigeic worm *Eisenia hortensis*. Epigeic worms live naturally in loose leaf litter and do not burrow. Yerkes' *E. foetida*, although smaller, is also epigeic. The worms can be maintained at normal room temperature. We used 8 of these worms in this experiment (ranging in weight from 0.06 – 0.64 g; mean to 0.40 g).

Worms were maintained in a commercially-available compost bin (Worm Factory 360, Nature's Footprint, Bellingham, WA, USA). They lived in a medium of coconut coir, shredded newspaper, and a bit of soil that was kept moist; vegetable scraps were added to this medium several times a week; worms should be considered to have had *ad libitum* access to food in their home environment. The temperature of the room housing the bin was held near 22°C; temperature within the bin rose as high as 29°C.

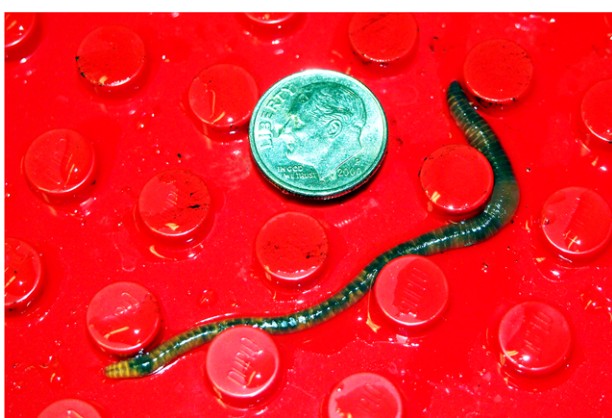

**Figure 1** **Earthworm on the Duplo board.** An earthworm is shown on the Duplo board. Note how the worm contacts the pegs that are near it. The US dime included for scale is 18 mm in diameter.

Earthworms are non-regulated animals, and therefore this research did not require the approval of our Institutional Animal Care and Use Committee.

### Apparatus

A Duplo base plate (15 × 15 in, (38 × 38 cm); Lego Corporation) with a rubber perimeter to create a barrier for the worm served as the behavioral apparatus. The Duplo base plate is smooth red plastic with raised circular pegs 4.5 mm high, 9 mm in diameter and positioned 16 mm from center to center. This provides an environment in which the thigmotaxic worm can feel things touching its body as if it were in soil, and the observer can still see the worm (see Fig. 1). A glass plate covered the Duplo plate. The Duplo plate rested on a 2 × 2 ft. (61 × 61 cm), $\frac{1}{2}$ − in− (1.25 cm)thick particle board supported at each corner by a 4 in (10 cm) wooden leg. (Note: British units given because they were the nominal measurements used to specify the purchased items.) A small electric motor (9–18 V, Radio Shack, 18,000 rpm) was attached to the board and served as the vibratory stimulus. A 205-lumen LED desk lamp rested on the board beside the Duplo plate, with its lighted face centered over the Duplo plate at a height of 24 cm, and positioned parallel to the plate; this resulted in a bright white light washing across the board, brightest in the board's center and less intense at the edges. Figure 2 illustrates the experimental apparatus. A 12 V automotive battery charger provided power for the motor and the lamp. Stimuli were controlled manually with the assistance of a stopwatch.

Water used to rinse the worm and to moisten the plate was tap water that had been sitting for several days. Experiments were conducted in a room illuminated by dim red light, to which earthworms are not sensitive (*Walton, 1927*).

### Procedure

Behavioral sessions were conducted between 1400 and 1600 h. Worms were selected from the bin and placed in a group in a small Styrofoam tub. Each session began with the worm being removed from the tub, placed on a paper towel, and sprayed lightly with water to

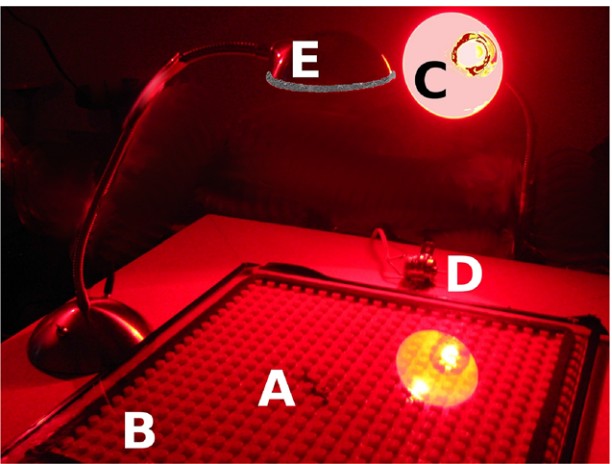

**Figure 2** **Our experimental apparatus.** An earthworm (A) is shown on our Duplo board (B). A red lamp (C) illuminates the board so that the worm can be observed. A small electric motor (D) serves as a Vibratory stimulus, and a bright white desk lamp (E) serves as the aversive Light stimulus.

remove any soil. The Duplo plate was sprayed with water, and the worm was transferred to the center of the plate with a thin wooden stick. A 15 min habituation period began when the glass cover was placed over the worm.

Experiment 1 involved the measurement of the worm's responses to the light and the vibration. Beginning 1 min after the habituation period, the worm received 18 trials at 2 min intervals. Three presentations each of six types of trials occurred: 3 Stimuli (Nothing, Light, or Vibration) for 2 Durations (10 or 30 s). Stimuli were presented pseudo-randomly, with no Stimulus occurring more than twice in a row, and no Duration occurring more than 3 times in a row.

Locomotion was determined by counting the movements of the worm's anterior end ("head") past the raised pegs on the Duplo plate. Every time the head crossed a "north-south" or "east-west" imaginary line connecting vertically- or horizontally-adjacent pegs a count was recorded. Movement could be forward or backward, but the line crossed had to differ from the line responsible for the previous count in order to be scored (thus a worm that retracted past a line then moved forward past that line would receive only a single count, until it crossed yet a different line). The movement score recorded for the worm was the mean of the responses to the three presentations of a trial type (e.g., Light for 10 s). Differences between the responses in the 3 conditions (No Stimulus, Light, & Vibration) and the 2 durations (10 s and 30 s) were assessed with a repeated measures analysis of variance. Effect sizes were determined by calculating a partial $\eta^2$ (*Tabachnick & Fidell, 2001*).

At the conclusion of the session the worm was retired to a different compost bin and not reused.
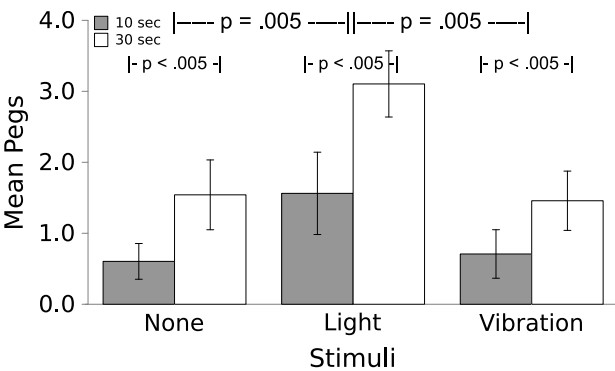

**Figure 3** **Movement in response to the Light and Vibratory stimulus.** Movement (mean ± S.E.) of the worms during 10- or 30 s periods during which No Stimulus, Light, or Vibration were presented. Worms moved significantly more to the Light than to the Vibratory stimulus (which caused no more movement than did the absence of a stimulus) and significantly more in the longer interval. There was no interaction.

## Results

Worms moved more in response to the Light than they did to the Vibration, which caused no change in locomotion over baseline (see Fig. 3; $F_{(2,7)} = 8.00$, $p = .005$); a partial $\eta^2$ of .533 indicates that this is a large effect. For both the Light and the Vibration, responses were greater for the 30 s duration than the 10 s duration ($F_{(1,7)} = 46.536$, $p < .005$); the effect size indicated by partial $\eta^2 = .869$ was very large. There was no Stimulus by Time interaction ($F_{(2,14)} = 0.83$, $p = .457$, partial $\eta^2 = .106$).

## Discussion

The results of Experiment 1 demonstrate that worms unconditionally respond to the Light stimulus by moving. Locomotor responses to the Vibratory stimulus were negligible, not differing from baseline movement. The $\eta^2 = .533$ suggests a medium to large effect, even with our small number of worms.

These results suggest that the Light stimulus might serve well as a US, causing an unlearned response of locomotion. This is not surprising; earthworms are vulnerable to predators if they are above ground in a lighted environment, and are also more likely to dry out, which would be fatal because they absorb oxygen from water on their surface (*Laverack, 1963*). The Vibratory stimulus was neutral with regard to locomotor responses, as others have shown (e.g., *Ratner & Miller, 1959*; *Herz, Peeke & Wyers, 1967*); this stimulus should serve well as a CS in Experiment 2. We expect that the movement elicited by the Light can be harnessed as an instrumental response for escape and avoidance learning.

## EXPERIMENT 2

Having determined that the earthworms move unconditionally to the bright Light stimulus, we set out in Experiment 2 to see if the worm's behavior could be brought under

stimulus control. Specifically, we paired the Vibration and Light, and designed an escape/avoidance procedure in which the worm could avoid the aversive Light stimulus by sufficient movement during the Vibration, or escape the Light by sufficient movement once the Light was presented. Because the mean movement during the 30 s Light in Experiment 1 was 3 pegs, we selected this as the amount of movement that would constitute either an avoidance or an escape response in this experiment.

## Materials & Methods

### Subjects

*Eisenia hortensis* were housed and cared for as in Experiment 1. 36 worms were used in this experiment, 18 in the Master (learning) group (0.28–1.16 g; mean = 0.55 g) and 18 in the Yoked (control) group (0.27–1.23 g; mean = 0.59 g).

### Apparatus

The same apparatus was used as in Experiment 1. In Experiment 2 the stimuli were controlled by computer interfaces and software (Med-PC, Med Associates, St. Albans, VT, USA).

### Procedure

Preparation of the worms and of the Duplo board matched those of Experiment 1. A 15 min habituation period commenced once the worm was placed on the board. Following this habituation the learning session occurred. During the learning session the experimenter observed the worm and pressed a switch that was monitored by the computer whenever the worm crossed a line between pegs, as defined in Experiment 1.

Our procedure is modeled after that of *Grau, Barstow & Joynes (1998)*. The Master group received 12 Escape/Avoidance trials (3 min ITIs, measured from CS-onset to CS-onset) in which a 30 s Vibratory CS predicted Light US onset. A movement of 3 pegs during the CS prevented the next US (avoidance R), but did not terminate the Vibratory CS; a movement of 3 pegs during the US resulted in its immediate termination (escape R). In the absence of an escape R, the US remained on for 30 s.

Each Control worm was yoked to a Master worm; it received the same number of Vibration and Light presentations, unpaired and in a pseudo-random order (each block of four trials included two Light and two Vibrations, randomly arranged). ITSs of 90 s resulted in a session of equal length to that of the Master Group, and assured that the intervals between successive Lights or Vibrations were comparable to those of the Master worms. Light durations were determined by the responses of a worm's yoked partner (e.g., if the 4th Light presentation for its yoked Master partner lasted only 17 s because an escape R occurred, then the 4th Light for this Control worm lasted 17 s; if the 7th Light was avoided by the yoked Master, then the 7th Light was not delivered to the Control worm).

Behavioral sessions for the Master worms began between 1013 and 2048 h (mean ± SE, 1400 h ± 58 min; 25th %ile = 1101, 75th %ile = 1543). Each Yoked worm by necessity began on average 75 min after its Master worm.

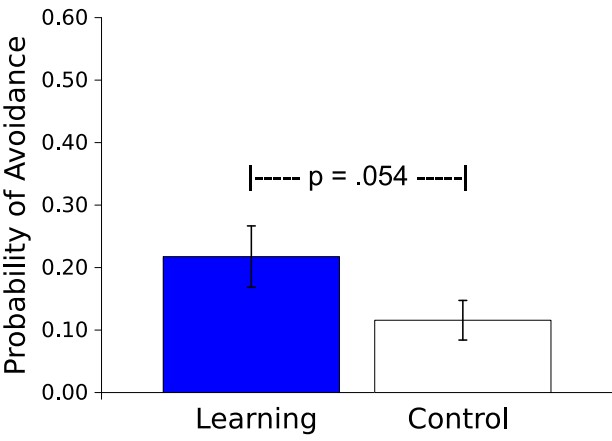

**Figure 4 Probability of an avoidance response.** Probability (mean ± S.E.) that a worm will respond during the 30 s presentation of the Vibratory stimulus. Worms in the Learning group, whose responses caused the subsequent Light not to be presented, responded more than did the yoked Control worms, whose responses had no effect.

For each worm we calculated the mean probability of responding during the Vibratory stimulus (avoidance response) and during the Light stimulus (escape response). Differences between the Master and Yoked worms were assessed by paired two-tailed t-tests. Effect sizes were assessed by calculating $\eta^2$ (*Tabachnick & Fidell, 2001*).

## Results

The mean probability of an avoidance response (defined as 3 peg crossings during the Vibratory CS) was greater for the Master worms than for the Controls ($t_{(17)} = 2.066$, $p = .054$, $\eta^2 = .201$; see Fig. 4). In the Master condition this avoidance response resulted in an absence of the US on that trial. For the yoked Control condition, US presentation was independent of the worm's movement, and was based wholly on the behavior of the corresponding Learning worm.

The probability of an escape response (defined as 3 peg crossings during the Light US) was greater for the Master worms than for the Controls ($t_{(17)} = 2.150$, $p = .046$, $\eta^2 = .214$; see Fig. 5). This escape response during the US by a Master worm caused US offset; behavior of the Control worm had no effect on the US, instead the duration of the US was determined by the behavior of the corresponding Master worm. As additional evidence of learning occurring during the session, in 13 of 18 dyads (72%) the Master worms made a higher proportion of their responses to the Light US in the second half of the Session than did the Control worms (Sign test, $p = .0245$, effect size of .692, *Grissom & Kim, 2012*).

## Discussion

The results of Experiment 2 demonstrated that the behavior of an earthworm can be affected by its consequences. In Master worms for which motor responses either prevented (Fig. 4) or eliminated (Fig. 5) an aversive Light stimulus those responses were more probable compared to the case in worms for which there was no contingency

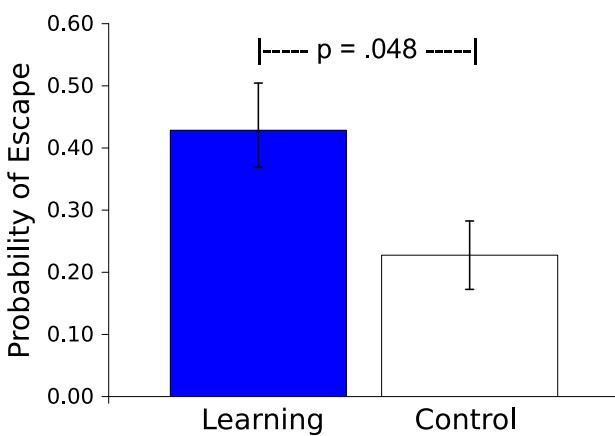

**Figure 5** **Probability of an escape response.** Probability (mean ± S.E.) that a worm will respond during the 30 s presentation of the Light stimulus. Worms in the Learning group, whose responses caused Light offset, responded more than did the yoked Control worms, whose responses had no effect.

between the responses and the Light. The $\eta^2$s suggest large effect sizes, indicating that the influence of contingency on behavior is strong. Thus worms are capable of avoidance and escape learning.

Comparisons across species are difficult, but it is worth noting that *Bower, Starr & Lazarovitz (1965)* demonstrated that with a procedure such as ours (i.e., when the avoidance response does not terminate the CS signaling the aversive stimulus) the probability of a locomotor avoidance response in rats is about.20 after 20 trials, comparable to our finding (Fig. 4). A further examination of how CS offset might affect this avoidance learning would be interesting, and we expect to pursue this. Furthermore, rats learn to escape shock at a rate comparable to what we saw in the earthworms: our Master and Yoked worms differed within the 12 trials of our training; *Campbell & Kraeling (1953)* showed that a response difference in running speed is apparent within as few as three trials between rats who escape shock by running and those who cannot escape.

## GENERAL DISCUSSION

We have demonstrated the feasibility of using locomotor activity of earthworms in response to Vibratory CSs and Light USs in studies of instrumental learning. In just one session we found that locomotor responses were more probable in worms for which prevention or offset of an aversive light stimulus was contingent on the behavior in comparison to worms for which there was no such contingency.

Escape responses occurred with a higher frequency than did avoidance responses, and there was a tendency for escape responses to increase over the course of the session in the Master group. It remains to be seen whether longer sessions or sessions repeated over several days will lead to more robust responding.

The use of the yoked control condition allows us to be sure that the differences in escape response probabilities were the result of the contingencies between responses and

stimuli rather than to any differences in exposure to the stimuli. The Light stimulus presentation and duration were contingent on behavior only for the Master worms. Because of the Vibration-Light pairing for the Master but not the Control worms, it is possible that the avoidance responses were Pavlovian conditioned responses rather than being instrumental in nature, although they had the effect of avoiding the Light. The present data do not allow an answer to this possibility; perhaps a procedure similar to that devised by *Izquierdo (1976)* to parcel out responses due to generalized drive, Pavlovian pairing, and instrumental contingency could provide an answer.

The differences that we observed were small; more work is necessary in order to demonstrate the rate at which escape and avoidance is acquired in the worm. We will examine the parameters of this learning in additional studies with repeated sessions, and we will assess the duration of the learned behavior by conducting tests at various times after the acquisition session. Basic learning phenomena such as generalization and discrimination should also be tested. Results of such studies will reveal the extent to which this learning in the earthworm is comparable to escape and avoidance conditioning in rodents and other species. Our ultimate goal is to establish an inexpensive model in which neural or chemical bases of learning can be examined easily.

### Funding

Some materials were provided by the Lego Corporation. Additional funding was provided by Albion College's Foundation for Undergraduate Research, Scholarship, and Creative Activity, Psychology Department, and Neuroscience Program. The funders had no role in study design, data collection and analysis, decision to publish, or preparation of the manuscript.

### Grant Disclosures

The following grant information was disclosed by the authors:
Lego Corporation.
Albion College's Foundation for Undergraduate Research, Scholarship, and Creative Activity, Psychology Department, and Neuroscience Program.

### Competing Interests

The authors declare that they have no competing interests.

### Author Contributions

- W. Jeffrey Wilson conceived and designed the experiments, performed the experiments, analyzed the data, contributed reagents/materials/analysis tools, wrote the paper.
- Nicole C. Ferrara and Amanda L. Blaker conceived and designed the experiments, performed the experiments, analyzed the data, wrote the paper.
- Charisa E. Giddings performed the experiments, analyzed the data, wrote the paper.

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
