# Peer review of "Escape and avoidance learning in the earthworm Eisenia hortensis"

_PeerJ, doi:10.7717/peerj.250_

## Round 0.1 · original submission · Minor Revisions

Please note that the formatting suggestions made by the reviewers are critical to the successful acceptance of this paper. Please take the time to adhere to threse suggestions completely.

Reviewer 1 ·

Basic reporting

Format/Figures:
Format:The format of the paper is that of a standard experimental psychology article which is a slight deviation from the basic PeerJ format, but is nicely formatted and easy to follow.

Introduction: Needs citation for Aristotle paragraph.
In the introduction the authors state: “Thus the earthworm might serve as a viable and inexpensive animal model for studies of the neural bases of learning.” It might be useful to point out that C.elegans serves this role in nematodes (ecdysozoans) and the earthworm may be the model in annelids (lophotrochozoans) so as to make clear that the investigators are aware of the existing model that is well-developed and has powerful genetics and non-invasive neural imaging methods.

Methods: Authors should address whether or not experiments were done at the same time of day since circadian effects on learning in E.fetida have been reported: P. Keshavamurthy, R.V. Krishnamoorthy, Behavioral Biology, Volume 20, Issue 1, May 1977, Pages 17–24.
Authors should address whether the experiments were always done within a certain period of time after feeding as time between feeding and expt has been shown to affect learning in C.elegans (a nematode and not an annelid, but the general concept might still be relevant). Here is the reference: EMBO J. 2011 Mar 16;30(6):1110-22. doi: 10.1038/emboj.2011.22. Epub 2011 Feb 8.
Food sensitizes C. elegans avoidance behaviours through acute dopamine signalling.
Ezcurra M, Tanizawa Y, Swoboda P, Schafer WR.


The figure legends need to be modified to meet the basic reporting standards. The figure legends should adhere to the following suggestion from PeerJ: “Figure legends should be self contained and clearly describe the figure and its contents.”
Therefore, the manuscript needs nearly complete revision of the figure legends. The current figure legends do not really describe the figures. A good figure legend should include a descriptive title sentence, a description of what is plotted or displayed, and a description of the error bars that does not require any additional calculations (i.e. S.D. or S.E.M. or otherwise). Information about methods should be placed in the methods section rather than the figure legend. Generally, the figure legend should not include interpretation or discussion of the data.

Experimental design

This seems fine, in general. The paper represents a new look at an old topic and the design is based on previous experimental psychology work, but with a new twist. I think the paper could be made more clear if the authors describe why a learning paradigm from rats was chosen (and not, say, from C.elegans or another annelid).

And, it might be useful to mention why n=8 and n=18 in expts 1 and 2, respectively. A simple power analysis likely suggests a larger sample size would be appropriate.

Validity of the findings

This is an interesting paper that might contribute nicely to the field of animal learning.

From an experimental design point of view the major concern about data validity is a low sample number (n=18) and whether there was observed a progressive learning over the 12 trials. The data for trial 12 are the only data shown. Was there a trend toward significance in the trials leading up to the 12th?

While I believe the data, I do wish the authors would simply state more clearly the specific statistical tests done. I gather they were one-tailed, T-tests. And the Eta-squared was calculated for the effect size. I think this should be stated very clearly in the methods. And, in multiple locations the reader is instructed on how to determine the probability for t-tests. It would be more appropriate for the authors to make a table including these values and reference them instead of having the reader do a calculation. In the very least I would suggest just placing this instruction 1 time in the methods and then presenting the one-tailed result since that is what the conclusions are based on. I guess you either have to believe the one-tailed result or not.

Last, the experimental apparatus is reasonable, but I do wonder if the increment of behavior that is measured is too crude. Perhaps a finer scale than the distance between pegs may have revealed a greater effect? I think that the effect may actually be under-reported with the current experimental approach--measuring peg crossing instead of actual worm displacement or something similar.

Additional comments

In addition to the revisions called for above I believe this paper could be made more clear if the following points are addressed:

Can the investigators be certain that the water sitting for several days was devoid of chlorine? Was the water tested? Might rephrase or remove.

How do the investigators know the worms do not respond to dim red light? Either need data showing that or a citation in the last sentence under Apparatus on p.5.

It’s hard to envision the light placement in their experimental apparatus and it’s not clear if there is a light gradient across the surface of the test arena.

Were worms ever reused? See last sentence under Procedure p.6.
Under Results on p.6 it is problematic to state worms moved “frequently” without some form of quantification. If this was not documented in the data acquired it might not be prudent to include here. If it was data recorded than a number should be put on this phenomena.

Under Apparatus, p. 7 the reference to Med Assoc could be rephrased with the company in parentheses…”…were controlled by computer software and interfaces (Med Assoc, Med-PC, City and State).

More liberal referencing of the data would help make this paper more clear. For instance, in the Discussion for Expt 2 the figure that is being discussed should be referenced.

·

Basic reporting

I would like to see a bit more comparison with rodent escape/avoidance literature to fully understand how the work fits into the broader field of knowledge

Experimental design

no comments

Validity of the findings

nice data!!

Additional comments

Overall this is a nice study of avoidance/escape learning in earthworms. The studies are well designed and are properly analyzed. Overall this short report is clear and well presented. Developing a good “model system” in which to study avoidance and escape is a very good idea and offers the opportunity to uncover cellular mechanisms of these behaviors. I believe the reader would be helped by a photograph of the worm in the apparatus (to get an understanding of the relative size of worms, board and pegs!!

In the general discussion could you compare the worm’s performance with that of a rodent? Is the worm faster/slower/about the same in rate of acquisition? Is the worm as smart as a rat? What should be tested next to advance this model? (Is there any information from rodent models that would guide the next steps?)

---

## Round 0.2 · accepted · Accept

Thank you for your updates. I am recommending to accept.